# The Improvement of the Irradiation Resistance of Amorphous MoS_2_ Films by Thermal Annealing

**DOI:** 10.3390/nano12030364

**Published:** 2022-01-24

**Authors:** Rui Zhang, Hong Zhang, Xiaoming Gao, Peng Wang

**Affiliations:** 1State Key Laboratory of Solid lubrication, Lanzhou Institute of Chemical Physics, Chinese Academy of Sciences, Lanzhou 730000, China; ruizhcn@gmail.com (R.Z.); zhanghong1220@licp.cas.cn (H.Z.); gaoxm@licp.cas.cn (X.G.); 2Center of Materials Science and Optoelectronics Engineering, University of Chinese Academy of Sciences, Beijing 100049, China

**Keywords:** Au^2+^ irradiation, amorphous MoS_2_ film, thermal annealing, irradiation resistance, lubricating properties

## Abstract

Among the structural materials used in fusion reactors, amorphous materials can effectively inhibit the accumulation and growth of radiation-induced defects, thereby improving irradiation resistance. However, the application of solid lubricating materials should also consider the changes in their lubricating properties after irradiation. This study shows that the ability to inhibit the deterioration of lubricating properties is not reflected in the amorphous MoS_2_ film. When the ion fluence reached 4.34 × 10^14^ ion/cm^2^, its wear life was reduced by two orders of magnitude, reaching 8.2 × 10^3^ revolutions. After the amorphous MoS_2_ film is vacuum annealed, its structural stability and resistance to deterioration of lubricating properties are improved. When the ion fluence reaches 1.09 × 10^15^ ion/cm^2^, for instance, the wear life of the MoS_2_ film annealed at 300 °C remains at 8.4 × 10^4^ revolutions. The higher irradiation tolerance of MoS_2_ films comes from the reduction in intrinsic defects by thermal annealing, which increases the internal grain size and volume fraction of grain boundaries, further providing an effective sink for irradiation defects.

## 1. Introduction

Nuclear fusion reactors must ensure the reliable operation of moving parts, as they have higher requirements for low friction and wear [1,2,3,4,5]. MoS_2_, as an excellent solid lubricant, is the preferred material for a nuclear fusion reactor. For example, the Wendelstein 7-X (W7-X) magnet system uses MoS_2_ film to ensure smooth and low-friction sliding of the two narrow support element (NSE) peers in the system under extreme operating conditions [3,4]. The MoS_2_-Ti-C composite film was deposited on the remote manipulator of Experimental Advanced Superconducting Tokamak to ensure its stability [5]. However, the solid lubricants used in nuclear fusion reactors suffer from the dual effects of friction wear and high-dose neutron irradiation [6,7]. Therefore, the irradiation resistance of the applied solid lubrication material is of great importance.

In our preliminary study, we found that pure MoS_2_ film has great limitations in its application, and its irradiation resistance is poor, meaning it was necessary to design its anti-irradiation structure [8,9]. Some studies have shown that amorphous structures lack long-range order and do not contain conventional lattice defects, thus offering the possibility of eliminating irradiation damage in polycrystalline solids [10,11,12,13,14,15]. For example, amorphous SiOC films deposited by magnetron sputtering showed no signs of crystallization, void formation or segregation in all irradiated samples [10]. In addition, interfaces such as grain boundaries inside the material can be used as “sinks” for irradiation point defects, thus reducing the irradiation damage of the material [16,17,18,19,20]. For example, a large volume fraction of grain boundaries in nanocrystalline Au can be used as sinks to absorb and eliminate point defects, thus showing better irradiation resistance than polycrystalline Au [18].

The internal structure of MoS_2_ films is affected by deposition methods and process parameters [21,22]. For example, a lower substrate temperature is favorable for the formation of amorphous structures [23]. In addition, thermal annealing is an effective method to reduce intrinsic defects and change the microstructure of films [24,25,26,27]. In a previous study, amorphous MoS_2_ films were prepared by controlling the deposition temperature and process parameters, and their internal structure was changed through vacuum thermal annealing [28]. In this work, the damage effects of amorphous and thermally annealed MoS_2_ films irradiated by Au^2+^ ions were compared. If neutron irradiation experiments are used to study the damage effects of amorphous MoS_2_ films, the research period is long and the experimental parameters are difficult to control. In addition, the radioactivity carried by the samples also brings difficulties to subsequent characterizations. Heavy ion irradiation can produce cascade damage similar to high-energy neutrons in the material, which can reach the irradiation fluence required by design in a short time. Moreover, the samples irradiated by heavy ions have almost no radioactivity, and the experimental parameters can be precisely controlled. The internal structure changes of the films before and after thermal annealing and irradiation were characterized by XRD and Raman spectroscopy. Then, the mechanical properties and lubricating properties of the MoS_2_ film were characterized. The structure and properties of amorphous and thermally annealed films before and after irradiation were compared. It was found that thermal annealing is an effective method to enhance the irradiation resistance of MoS_2_ films.

## 2. Materials and Methods

### 2.1. Material Preparation

The amorphous MoS_2_ film was deposited by a closed-field unbalanced-magnetron sputtering (CFUBMS) system (HTC-1200, Hauzer Corp, Venlo, Netherlands) using a rectangular pure MoS_2_ target on Si wafers (for characterizing wear behavior and other analysis) [28]. The deposition was implemented at argon pressure of 1.5 Pa, 4 kW sputtering power and −50 V bias voltage. After deposition, amorphous MoS_2_ films were thermally annealed at 300 and 500 °C for 3 h in a vacuum environment and then cooled naturally to room temperature.

### 2.2. Heavy Ion Irradiation

Heavy ion irradiation experiments were performed on 1.7 MeV tandem accelerators in the ion beam materials laboratory (IBML) at China. Before the irradiation experiment, the calculation mode of ion distribution and quick calculation of damage was adopted by SRIM to simulate the displacement damage depth and Au ion distribution of MoS2 film at an irradiation fluence of 1.09 × 10^15^ ion/cm^2^ [29]. The films were irradiated with 2 MeV Au^2+^ ions at ion fluences of 4.34 × 10^14^ and 1.09 × 10^15^ ion/cm^2^, which corresponded to displacement per atom (dpa) of 2 and 5, respectively. The relationship between the fluence of incident Au ions and displacement per atom was correlated by the following formula [30]:(1)$$dpa=\frac{N_{displacement}\Phi }{N_{W}}$$

Φ: Implantation fluence, ion/cm^2^

*N_displacement_*: Number of displacement, 10^7^/(ion·cm)

*N_W_*: Tungsten atomic density, atom/cm^3^

The experiments were achieved at room temperature. The Au^2+^ ions were accelerated in an accelerating vessel with a high-voltage electric field (1.7 tandem accelerator). During irradiation, the incident angle of the Au^2+^ beams was perpendicular to the sample surface, and the current of the incidence beams ranged from 100 to 200 nA. Because the current intensity of the incident ion beam is relatively weak, the temperature of the sample during irradiation is 10–20 °C higher than room temperature. This has little effect on the experimental results.

### 2.3. Structural and Properties Characterization

The crystal structure of the films was characterized by grazing incidence X-ray diffraction (GIXRD, D8 Advance, Bruker, Karlsruhe, Germany) with Cu Kα radiation, and the diffractograms were acquired from 10° to 70°. A Horiba LabRam HR800 Micro-Raman Spectroscopy and X-ray energy dispersive spectrometer (EDS) were applied to measure the composition. HRTEM (TECNAI G2 S-TWIN F20, FEI, Hillsboro, OR, USA; accelerating voltage, 200 kV) was carried out to observe the cross-sectional thickness and microstructures of films. The hardness of amorphous and thermally annealed MoS_2_ films before and after ion irradiation was obtained using a nanoindenter (TI950, Hysitron TriboIndenter, Minneapolis, MN, USA) with a Berkowich diamond tip. The indentation depth was controlled to ~200 nm to exclude the influence of substrates. The lubricating properties of the films were evaluated using a homemade ball-on-disk tribometer in a vacuum environment (≤5.0 × 10^−4^ Pa) under a normal load of 3.0 N and a rotational speed of 1000 r/min. The disk was the MoS_2_-deposited Si substrate and the counterpart was AISI 440C steel balls (with a diameter of 8 mm and Ra ≤0.1 μm).

## 3. Results

XRD was used to characterize the crystal structure changes of amorphous and thermally annealed MoS_2_ films before and after irradiation. The test spectrum is shown in Figure 1. In Figure 1b, a small number of short-range ordered structures exist in the original amorphous MoS_2_ film. After vacuum thermal annealing, it can be seen that the intensity of the (002) diffraction peak of the MoS_2_ film is significantly increased. Thermal annealing causes the recrystallization of the film and promotes the growth of the original short-range structure and the nucleation of new crystal grains. The thermally annealed MoS_2_ film is polycrystalline and grows in the preferred orientation (002) parallel to the bottom surface (300 °C, 500 °C: 2θ = 13.55°), which is beneficial to the lubricating properties of the MoS_2_ film [31,32]. In addition, there are (100) and (103) diffraction peaks (300 °C: 2θ = 33.75°, 39.25°; 500 °C: 2θ = 33.65°, 39.75°, respectively). These two wide diffraction peaks indicate that the orientation has poor crystallinity or that there is only the short-range ordered structure.

The internal structure of the amorphous MoS_2_ film is more disordered, and the short-range ordered structure inside the film is destroyed after the irradiation experiment. When the irradiation damage level increases to 5 dpa, the internal structure of the film is in a state of complete disorder. For the comparison of XRD patterns of thermally annealed films before and after irradiation, it can be found that when the irradiation damage level reaches 5 dpa, (002) and (100) diffraction peaks still exist. This shows that thermally annealed films still retain some grains after irradiation. The (002) and (100) peak intensities of the annealed film were normalized, and the results are shown in Figure 2. The histogram visually shows the changes in diffraction peak intensity before and after irradiation. When the irradiation damage level reached 5 dpa, the intensity variations of the (002) or (100) diffraction peaks caused by the irradiation of the 300 °C-annealed film is less than that of 500 °C-annealed film. Moreover, the stability of the (100)-oriented crystal is better than that of the (002)-oriented crystal based on the changes in the two diffraction peaks. However, for amorphous films, its diffraction peaks are typical broad peaks. There are a small number of short-range ordered structure inside, which have little effect on the intensity of diffraction peaks. Therefore, after the irradiation experiment, the change in the diffraction peak intensity of the amorphous film is not obvious.

The element content of the film before and after thermal annealing was tested by EDS, and the results are shown in Table 1. Due to the loss of S, the S/Mo ratio is less than the ideal stoichiometric ratio, and the film is mainly composed of MoS_2_ and MoS_x_ [33]. The presence of O also causes the film to contain a certain amount of MoO_2_ and MoO_3_ [34]. In the process of thermal annealing, thermal annealing will make MoO_3_ unstable and then transform into MoO_2_ [28]. The comparison of the element contents in the three films shows that thermal annealing increases the S/Mo ratio of the films, and the content of O decreases with increasing annealing temperature. The samples inevitably adsorb moisture and oxygen in the air during storage and testing, thereby increasing the oxygen concentration on the surface of the samples. When the samples are thermally annealed in vacuum, under the effect of low pressure (10^−4^ Pa) and a temperature field, part of the moisture and oxygen desorbs from the surface of the samples and is pumped out of the chamber by the molecular pump, which leads to a decrease in the oxygen concentration on the surface of the annealed samples.

Raman spectroscopy was used to further study the chemical structure near the surface of the film before and after thermal annealing and irradiation, as shown in Figure 3. The spectra show the Raman peak changes before and after thermal annealing or irradiation, and demonstrate that the Raman peak changes are the same after irradiation for all samples. Combined with the XRD results, thermal annealing causes recrystallization of the amorphous MoS_2_ film, so the Raman spectra at 378 and 405 cm^−1^ clearly show two typical MoS_2_ crystal vibration peaks: E^1^_2g_ and A^1^_g_. With increasing annealing temperature, the crystal integrity in the film increases, and the intensity of the corresponding Raman vibration peak increases. There are two vibration peaks at 378 and 405 cm^−1^ in the thermally annealed film before irradiation. When the irradiation amount reached 4.34 × 10^14^ ion/cm^2^, the peak intensities of E^1^_2g_ and A^1^_g_ disappeared, and the irradiation fluence increased further, with the same results. It can be seen in XRD that the film retains MoS_2_ crystals after irradiation. Considering the surface detection depth and sensitivity of the Raman spectrum to the surface structure, the disappearance of the Raman peak can be attributed to the severe destruction of the MoS_2_ crystal on the film surface under heavy ion bombardment [35,36,37].

According to the above test results, in order to further clarify the influence of heavy ion irradiation on the film, the actual depth of influence of the ion irradiation inside the film was tested by HRTEM, as shown in Figure 4. The figure indicates that there is a clear dividing line between the irradiated area and the non-irradiated area. The thickness of the film is about 2.48 μm, and the influence depth of incident ions in the film is about 0.66 μm, accounting for 26.6% of the film thickness. The influence depth of the irradiated ions is related to the film categories and the energy of the incident ions, so the three films have the same influence depth [29].

Likewise, to more intuitively show the changes in the internal microstructure of amorphous MoS_2_ film before and after thermal annealing, HRTEM was used to characterize the cross-section of the film, and the results are shown in Figure 5. The interior of the amorphous MoS_2_ films is in a dense disorder state. After the film is annealed at 300 °C, the thermal effect causes the grains to nucleate and grow, but there are still amorphous regions inside the film, forming an amorphous/nanocrystalline composite structure. After the film is annealed at 500 °C, there is a crystalline phase structure inside the film, and almost no amorphous region exists.

The mechanical and tribological properties of the films before and after thermal annealing and irradiation were tested. The hardness changes of the three kinds of films before and after irradiation are shown in Figure 6. The indentation depth of the experiment was 200 nm to produce enough plastic deformation and to avoid the substrate effect during the indentation experiment. The mechanical properties of the films before irradiation showed that the hardness of amorphous MoS_2_ films increased with increasing thermal annealing temperature (1.08–1.24–1.33 GPa). This hardness change is related to the microstructure of the film. As shown in Figure 1, Figure 2 and Figure 3, after the film is annealed, a large amount of crystal is formed, which increases the volume fraction of grain boundaries inside the film. The transition from disorder to order in MoS_2_ also makes the internal structure of the film dense [28]. The large volume fraction of grain boundaries and densified structure can inhibit the formation and propagation of cracks during plastic deformation, thereby improving the hardness and load-bearing capacity of the film. When the irradiation damage level reaches 2 dpa, the hardness of the irradiated film is more than three times that of the as-deposited film, showing weak resistance to irradiation hardening. The irradiation hardening caused by irradiation exceeds the acceptable limits of the amorphous MoS_2_ film, making the film brittle. This is not conducive to its lubrication performance and load resistance performance.

Annealing improves the mechanical properties of the film, and its hardness changes within a suitable range (1.08–1.24–1.33 GPa), which is also conducive to the improvement of its lubricating properties. When the thermally annealed film is irradiated, the samples annealed at 300 and 500 °C show different irradiation-hardening trends, but their irradiation-hardening resistance is better than that of the as-deposited films. The hardness of the film annealed at 300 °C increases gradually with an increasing irradiation damage level. The hardness of the film annealed at 300 °C is 1.24 GPa, and when the irradiation damage level is increased to 2 dpa, the hardness is 1.95 GPa, an increase of 57.3%. When the irradiation damage level was further increased to 5 dpa, the hardness was 2.98 GPa, increasing by 52.8%. The hardness of the films annealed at 500 °C is 1.33 GPa, and when the irradiation damage level reaches 2 dpa, the hardness is 2.59 GPa, increasing by 94.7% and approaching the saturation state. When the irradiation damage level further increased to 5 dpa, the hardness was 2.66 GPa, which increased by only 2.7%. One reason for the increase in hardness is due to the formation of point defects in MoS_2_ film by ion bombardment, which form vacancy groups and gap clusters after recombination, aggregation and evolution, causing lattice distortion and hinders dislocation movement, thus forming irradiation hardening [38]. Another reason is the compact amorphization caused by irradiation-induced displacement of lattice atoms [39]. These two reasons correspond to the crystalline and amorphous regions after irradiation, respectively. The hysteresis loop of the film is statistically analyzed, and the results are shown in Figure 7. The test results show that the plastic deformation resistance of the amorphous MoS_2_ film is improved (28.1–42.8–55.7%) after thermal annealing. After irradiation, the plastic deformation resistance of the three kinds of film is greatly improved, and the increase in thermal annealing films is more obvious.

The influence of irradiation on the internal structure and mechanical properties of the film will lead to a change in its lubricating properties. Figure 8 shows a comparison of the wear lives of amorphous and thermally annealed MoS_2_ films before and after irradiation. Figure 8a shows that the wear life of the as-deposited film is 16.7 × 10^4^ r, and the wear lives of the film after thermal annealing at 300 and 500 °C increase to 23.6 × 10^4^ r (an increase of 41.3%) and 25.7 × 10^4^ r (an increase of 53.9%), respectively. This shows that the formation of MoS_2_ crystals is beneficial to improve the lubricating properties of the film. After the irradiation experiment, the wear life of the film is significantly reduced due to damage to the internal crystal structure and the brittleness problem caused by the large increase in hardness, especially the wear life of the amorphous MoS_2_ film. When the irradiation damage level reaches 2 dpa, the wear lives of the as-deposited and annealed films decrease by 95.2%, 49.6% and 79.4%, respectively. When the irradiation damage level increases to 5 dpa, the wear lives of the amorphous and annealed films are 8.2 ×10^3^ r, 83.6 ×10^3^ r and 27.5 ×10^3^ r, respectively. From the change in wear lives, the irradiation resistance of the amorphous MoS_2_ film increases most obviously after being annealed at 300 °C. Observing the trend of the curve, it can be found that the friction process of the annealed MoS_2_ film is more stable and the friction coefficient decreases. After irradiation, the friction process of the film is unstable, and the friction coefficient fluctuates and rises.

As shown in Figure 9, Raman spectra were obtained in the central and edge regions of the wear track of amorphous and thermally annealed films before and after irradiation. The comparison of the data of the pristine sample shows that the order degree of the friction-induced ordering film in the center of the wear tracks of as-deposited film is higher than that of the annealed films, and the order degree in the center of wear tracks is higher than that of the edge for all samples. It can be seen that the higher the crystallinity of the film is, the less conducive it is to the formation of friction-induced ordered films. After the Au^2+^ irradiation experiment, the order degree of the induced film in the wear track center of all films is lower than that of the edge. This shows that all irradiated films have the same behavior in the friction process: in the friction process, the friction-induced ordered film with a high degree of order is pushed from the center to the edge. This behavior leads to a decrease in the wear life of the irradiated film. Similarly, with increasing crystallinity, the order degree of the friction-induced ordered film decreases, so the wear life of the 300 °C-annealed film is higher than that of the 500 °C-annealed film. The absence of the induced ordered film is the main reason for the increase in the friction coefficient of the irradiated film.

In summary, amorphous MoS_2_ film does not show anti-irradiation performance in tests of mechanical properties and lubricating properties. After vacuum thermal annealing, the irradiation-hardening resistance of annealed films is significantly better than that of amorphous films (increases by 16.3% for 300 °C-annealed film and increased by 25.3% for 500 °C-annealed film). The wear life test of films shows the change in irradiation resistance more directly. After irradiation, the wear life reduction of the film annealed at 300 °C is smallest, which is 35.4% of the pristine life (irradiation damage level is 5 dpa). After thermal annealing, the short-range ordered structure in the amorphous MoS_2_ film grows further under the action of the thermal effect, and promotes the nucleation of new grains. With increasing grain size, the grain boundaries become increasingly obvious. The thermal effect causes the disordered atoms to rearrange at the low-energy interface, thus reducing the internal energy of the film and improving the structural stability and irradiation resistance. The XRD results show that the crystal structure stability of thermally annealed films is obviously improved. In terms of why the anti-irradiation performance of the film annealed at 500 °C is decreased, combining XRD and Raman data, it can be concluded that the film annealed at 300 °C is an amorphous/nanocrystalline composite structure, while the film annealed at 500 °C is a crystalline structure, and the former has stronger anti-irradiation performance. First, studies have shown that the smaller the grain size is, the higher the sink intensity, and the easier it is to absorb irradiation defects [40]. Second, compared with normal crystalline and amorphous materials, the internal interface of amorphous/nanocrystalline composite structure materials has been proven to be an effective “sink” for point defects [41,42]. The grain boundaries in crystal materials tend to absorb interstitial atoms, while vacancies caused by irradiation in composite structure film materials can also annihilate rapidly and completely at the boundary between nanograins and adjacent amorphous regions, thus inhibiting the aggregation of vacancies and interstitial atoms and greatly reducing the radiation damage of materials [43]. Therefore, the 300 °C-annealed film shows better irradiation resistance.

## 4. Conclusions

In this study, the amorphous MoS_2_ film did not exhibit irradiation resistance, and its irradiation resistance was greatly increased after vacuum thermal annealing. After thermal annealing, the original short-range ordered structure of the film grows further, and new crystal grains begin to nucleate. The mechanical properties and lubricating properties of the film are improved with increasing annealing temperature (hardness: 1.08–1.24–1.33 GPa; Wear life: 16.7 × 10^4^–23.6 × 10^4^–25.7 × 10^4^ r). After the irradiation experiment, the annealed film shows better resistance to irradiation hardening and structural stability. Among them, the film annealed at 300 °C has the best irradiation resistance. Compared with the film annealed at 500 °C, the film annealed at 300 °C shows higher sink strength, stronger structural stability and lower wear life reduction in the irradiation process. This study shows that thermal annealing is an effective method to enhance the irradiation resistance of amorphous MoS_2_ films. An amorphous/nanocrystalline composite structure with strong irradiation resistance can be formed in the film at the proper annealing temperature.

## Figures and Tables

**Figure 1 nanomaterials-12-00364-f001:**
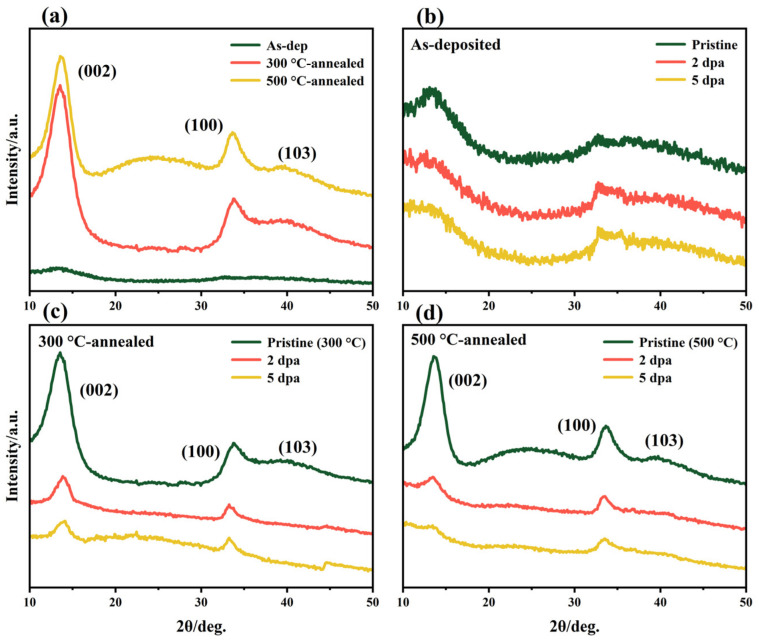
XRD patterns of the as-deposited film and thermally annealed films: (**a**) amorphous MoS_2_ film before and after thermal annealing; (**b**) as-deposited, (**c**) 300 °C-annealed and (**d**) 500 °C-annealed film before and after irradiation.

**Figure 2 nanomaterials-12-00364-f002:**
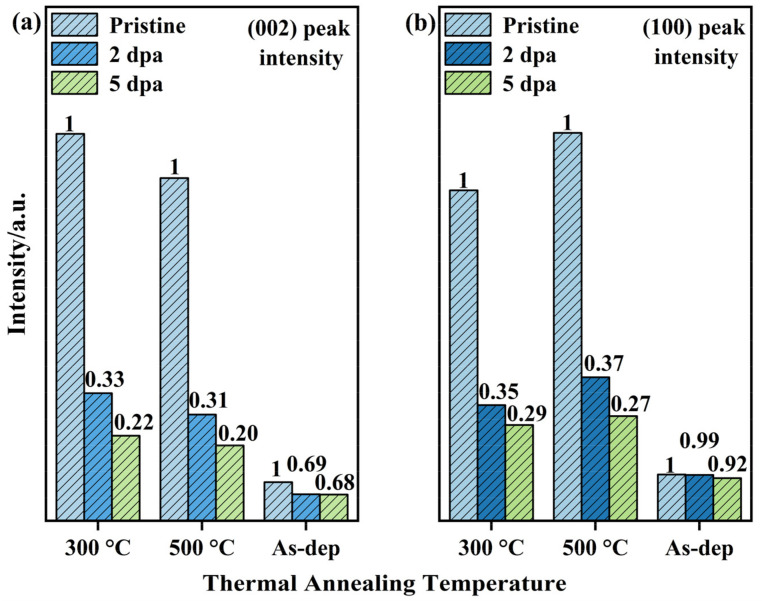
The (002) and (100) peak intensity variations of the XRD diffraction peaks of the thermally annealed films before and after irradiation (all samples were processed by the normalization method). In addition, the amorphous MoS_2_ film has no obvious diffraction peak, and its data are the intensities of broad peaks at 2θ = 13.78° and 33.15°: (**a**) (002) peak intensity and (**b**) (100) peak intensity.

**Figure 3 nanomaterials-12-00364-f003:**
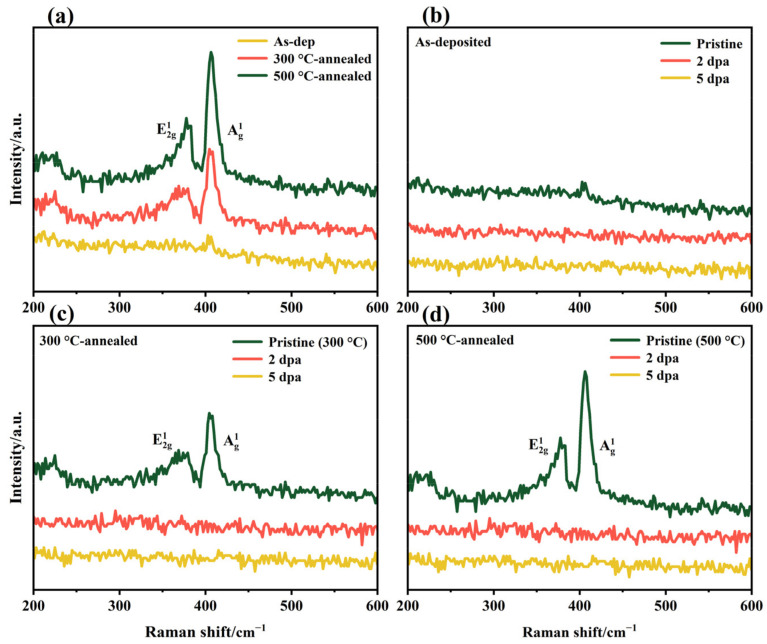
Raman spectra of the as-deposited film and thermally annealed films: (**a**) amorphous MoS_2_ film before and after thermal annealing; (**b**) as-deposited, (**c**) 300 °C-annealed and (**d**) 500 °C-annealed film before and after irradiation.

**Figure 4 nanomaterials-12-00364-f004:**
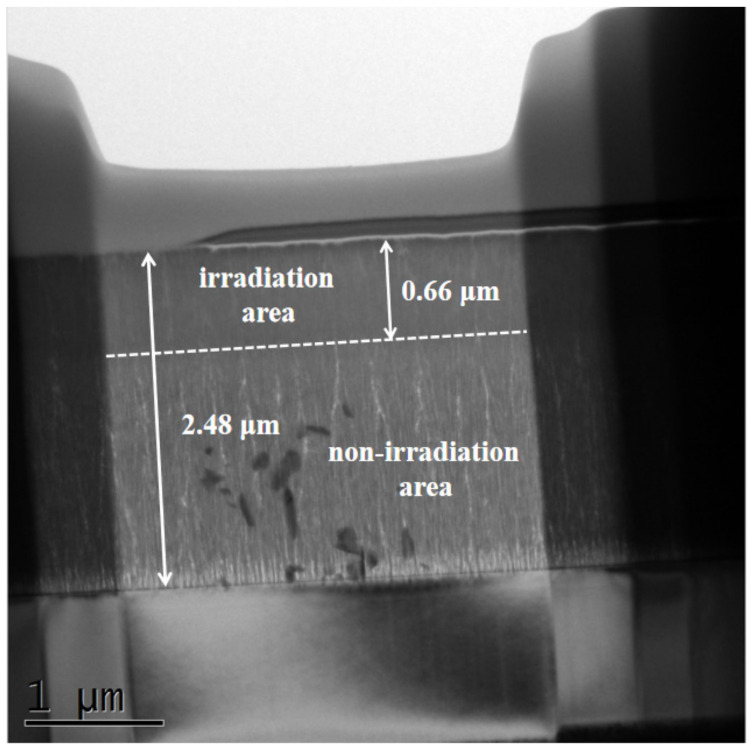
Cross-sectional HRTEM image of the as-deposited film at a damage level of 5 dpa.

**Figure 5 nanomaterials-12-00364-f005:**
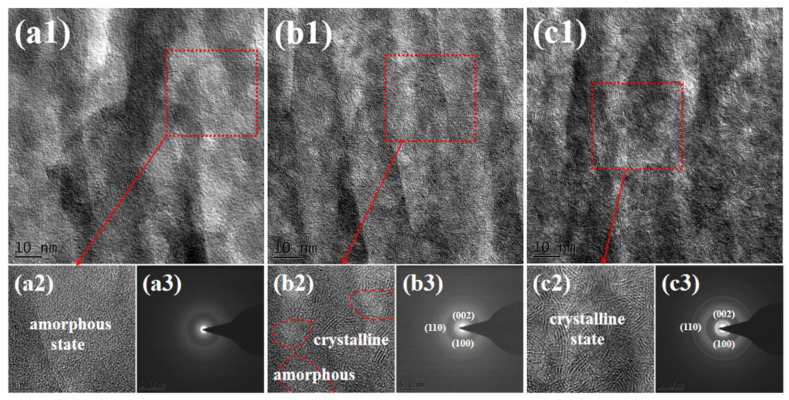
Cross-sectional HRTEM images of the as-deposited and thermally annealed films (including TEM images and SAED patterns): (**a1**–**a3**) as-deposited, (**b1**–**b3**) 300 °C-annealed and (**c1**–**c3**) 500 °C-annealed films.

**Figure 6 nanomaterials-12-00364-f006:**
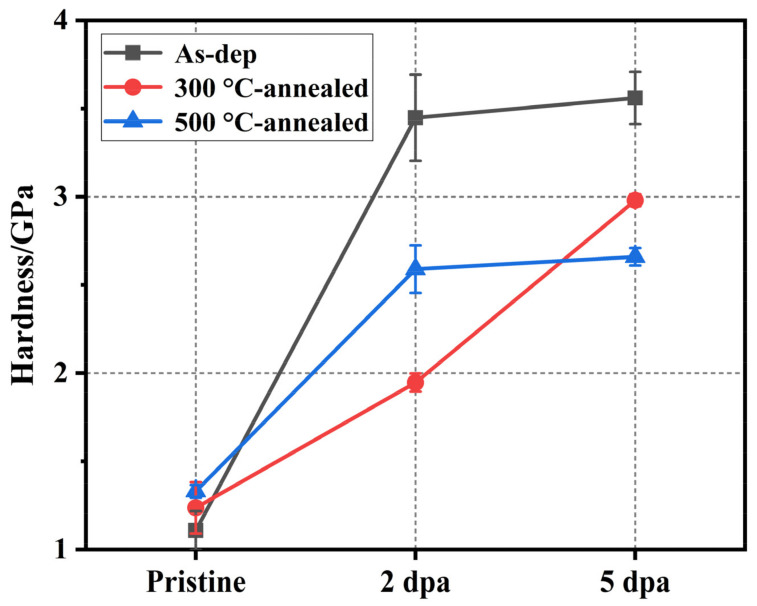
Hardness of the as-deposited film and thermally annealed films (300 °C-annealed and 500 °C-annealed) before and after ion irradiation.

**Figure 7 nanomaterials-12-00364-f007:**
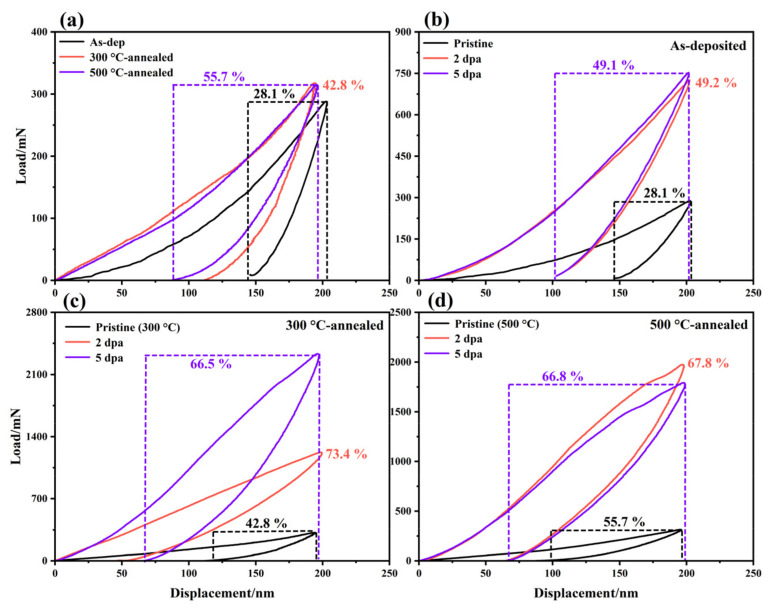
Hysteresis loop of the as-deposited film and thermally annealed films: (**a**) amorphous MoS_2_ film before and after thermal annealing; (**b**) as-deposited, (**c**) 300 °C-annealed and (**d**) 500 °C-annealed film before and after irradiation.

**Figure 8 nanomaterials-12-00364-f008:**
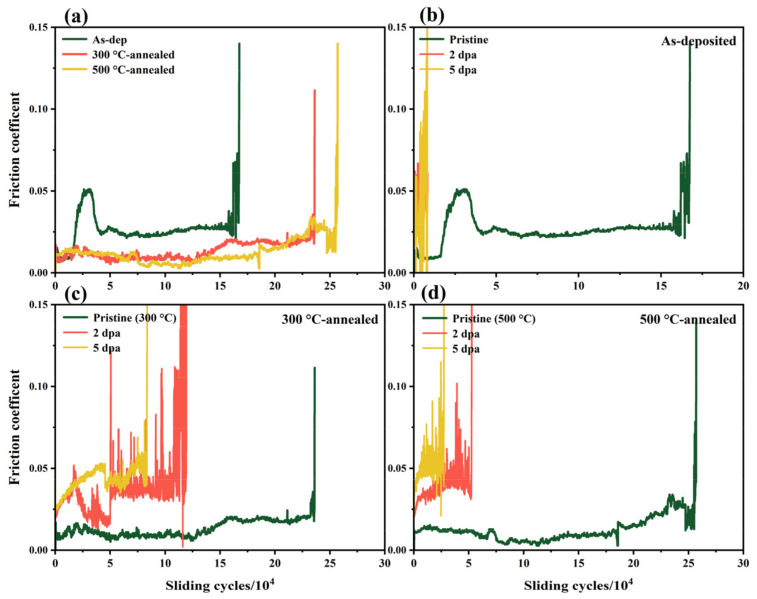
The wear lives of the as-deposited film and thermally annealed films: (**a**) amorphous MoS_2_ film before and after thermal annealing; (**b**) as-deposited, (**c**) 300 °C-annealed and (**d**) 500 °C-annealed film before and after irradiation.

**Figure 9 nanomaterials-12-00364-f009:**
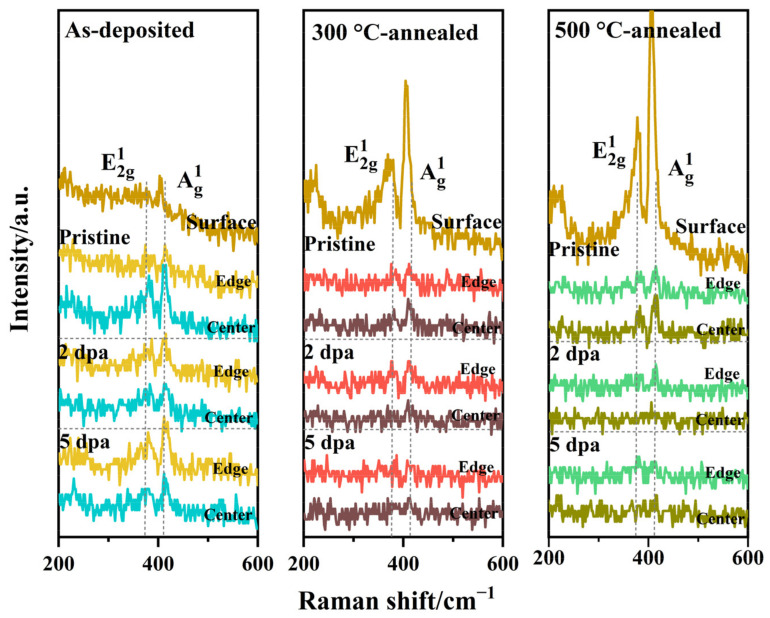
Raman spectra in the wear track (center and edge) of the as-deposited (**a**) and thermally annealed films (300 °C-annealed (**b**) and 500 °C-annealed (**c**)) before and after irradiation after the friction test.

**Table 1 nanomaterials-12-00364-t001:** Chemical compositions of amorphous and thermally annealed MoS_2_ films.

Sample	Temperature (°C)	Chemical Composition (at. %)			
		Mo	S	O	S/Mo
MoS_2_	--	30.50	50.36	19.14	1.65
Annealed MoS_2_	300	31.01	52.99	16.00	1.71
Annealed MoS_2_	500	30.34	55.13	14.53	1.82

## Data Availability

The data that support the findings of this study are available from the corresponding author upon reasonable request.

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
