# Peer review of "The Improvement of the Irradiation Resistance of Amorphous MoS2 Films by Thermal Annealing"

_nanomaterials, 2022, doi:10.3390/nano12030364_

Round 1
Reviewer 1 Report
This paper describes very well how to enhance the irradiation resistance of amorphous MoS2 films by thermal annealing. The quality of the presentation is very good. I suggest only to modify the abstract because the first three sentences are not clear or correct in english.
Although the amorphous structure can effectively inhibit the aggregation and growth of the irradiation induced defects, and enhance the irradiation resistance of the materials.
Rewrite. I don't find the principal
However, this study shows that amorphous MoS2 film has the poor irradiation resistance and serious lubrication degradation, its wear duration reduced by two orders of magnitude to 8.2 ×103 r after ion irra-14 diation up to the damage level of 2 dpa.
English check please.
After vacuum heat treatment, although the hardness of the annealed film increases after irradiation but it shows a longer wear duration and better tribological performance.
English check please.
Author Response
Thank you very much for your suggestions and questions. We have modified these three sentences as follows:
“(1) Among the structural materials used in fusion reactors, amorphous materials can effectively inhibit the accumulation and growth of radiation-induced defects, thereby improving its radiation resistance. However, the application of solid lubricating materials should also consider the changes in their lubricating properties after irradiation. (2) This study shows that the ability to inhibit the deterioration of lubricating properties is not reflected in the amorphous MoS2 film. When the ion irradiation damage level up to 2 dpa, its wear live is reduced by two orders of magnitude, reaching 8.2 × 103 rotates. (3) After the amorphous MoS2 film is vacuum annealed, its structural stability and resistance to deterioration of lubricating properties are improved.”
Reviewer 2 Report
This manuscript contains full of contradictions between data. For one example, irradiation causes hardness increase, but the crystal structure collapses. However, hardness increases as the crystal structure develops based on the heat-treated films before irradiation.
Furthermore, English is poor, contains wrong terminologies and ambiguous words. The introduction and results does not match. The introduction explains that lubrication property of MoS2 is important in fusion reactor which is subjected to neutron irradiation. However, the experiment was conducted with Au ion. Despite the same dpa, neutron and heavy charged particle irradiation shows different effects on material.
Above all, it is difficult to understand this study. Based on the results, irradiation hardening should negatively affect the lubricating property of MoS2. The amorphous MoS2 film is better against irradiation in this point of view. However, the title is "Enhance the irradiation resistance of amorphous MoS2 films by thermal annealing". The term irradiation resistance seems to be "suppression of the deterioration of lubricating properties of MoS2 film subjected to irradiation", but the results indicate the opposite.
Author Response
Thank you very much for your questions.
- From the data graph of film’s hardness, it can be found that the hardness of the MoS2 film increases with the increase of the annealing temperature, but the amount of change is small. This shows that the crystal structure develops based on the heat-treated film before irradiation has a limited effect on the hardness of the MoS2 film. After the irradiation experiment, the irradiation hardening phenomenon of the film is particularly obvious. This shows that, as described in the article, the densified amorphous layer (internal energy increase) and lattice distortion caused by irradiation damage have a more significant impact on the hardness of the MoS2 film.
- If neutron irradiation experiments are used to study the damage effects of amorphous MoS2 films, the research period is long and the experimental parameters are difficult to control. In addition, the radioactivity carried by the samples also brings difficulties to subsequent characterizations. The heavy ion irradiation can produce cascade damage similar to high-energy neutrons in the material, which can reach the irradiation fluence required by design in a short time. Moreover, the samples irradiated by heavy ions have almost no radioactivity, and its experimental parameters can be precisely controlled.
- I am not quite sure what you mean. What you said "Based on the results, irradiation hardening should negatively affect the lubricating property of the MoS2" is correct. Because the large increase in hardness will make the film brittle, which is not conducive to its lubricating properties. From this point of view, the amorphous MoS2 film in this experiment has the worst irradiation resistance rather than the better. The irradiation resistance and anti-lubricating properties degradation of the MoS2 film have similarities. The stability of the internal structure of the film can preserve its lubricity to a greater extent. The results of this study are also the same, the ability of the treated film to resist irradiation hardening and anti-lubricity deterioration is improved compared with the amorphous film.
Reviewer 3 Report
The article is devoted to the study of the effect of thermal annealing on the radiation resistance of MoS2 films. In general, this direction is of scientific interest and practical significance, since obtaining new data in the field of resistance of materials to radiation is one of the priority areas of research in modern materials science. The very direction presented in the work is of interest not only for a wide range of readers, but also for specialists in the field of studying the kinetics of radiation damage. In the opinion of the reviewer, the article can be accepted for publication after the authors make corrections and answer all the questions that arise when reading it.
1. The authors should provide a method for calculating the displacements per atom, for the given values ​​in the work. This is an important fact, since in most works this value is determined incorrectly, which leads to a discrepancy in the results.
2. Regarding irradiation, was the temperature of the samples measured during irradiation in order to avoid thermal heating of the samples and annealing of radiation defects, respectively?
3. The authors point to the processes of amorphization as a result of irradiation, however, from the presented diffraction patterns it is not entirely clear how exactly the degree of amorphization of the films was estimated, was this value determined numerically, or was it just an observed effect determined by the change in the intensity of the main diffraction reflections?
4. What is the reason for the change in the oxygen concentration during annealing of the samples?
5. The presented results of changes in hardness require a different interpretation or presentation, in this form they raise questions about the effect of annealing on increasing load resistance.
Author Response
Thank you very much for your suggestions and questions.
- The relationship between the fluence of incident Au ion and displacement per atom were correlated by the following formula:
(The formula cannot be written, please see the attachment. Sorry) (1)
Ф: Fluence, ion/cm2
Damage-rate: Vacancies, ion-Å
Natom: Atomic density, atom/cm3
- The temperature of the sample is measured during the irradiation process. Because the current intensity of the incident ion beam is relatively weak, the temperature of the sample during irradiation is 10-20 °C higher than room temperature. This has little effect on the experimental results.
- The degree of amorphization of the film cannot be realized with intuitive numbers. In this experiment, we can confirm that as the irradiation fluence increases, the degree of amorphization becomes higher and higher. So we use another angle (the change in the intensity of the crystal diffraction peak) to determine the extent to which the internal crystal structure of the film is destroyed by heavy ions. The higher the degree of the crystal from order to disorder, the lower the intensity of diffraction peaks. But for amorphous films, its diffraction peaks are typical broad peaks. There is a small amount of short-range ordered structure inside, which has little effect on the intensity of diffraction peaks. Therefore, after the irradiation experiment, the change of the diffraction peak intensity of the amorphous film is not obvious.
- The samples inevitably adsorb moisture and oxygen in the air during storage and testing, thereby increasing the oxygen concentration on the surface of the samples. When the samples are thermally annealed in vacuum, under the effect of low pressure (10-4 Pa) and temperature field, part of the moisture and oxygen desorbs from the surface of the samples and is pumped out of the chamber by the molecular pump, which leads to a decrease in the oxygen concentration on the surface of the samples after annealing.
- After the film is annealed, a large amount of crystal is formed, which increases the density of grain boundaries inside the film. The transition from disorder to order of MoS2 also makes the internal structure of the film dense [28]. The high-density grain boundaries and densified structure can inhibit the formation and propagation of cracks during plastic deformation, thereby improving the hardness and load-bearing capacity of the film. Annealing improves the mechanical properties of the film, and its hardness changes within a suitable range (1.08-1.24-1.33 GPa), which is also conducive to its lubricating properties. The radiation hardening caused by radiation doubles the hardness of the film, which exceeds the endurance limit of the film, which makes the film brittle. This is not conducive to its lubrication performance and load resistance performance.
([28] Zhang, R.; Cui, Q.; Weng, L.; Sun, J.; Hu, M.; Fu, Y.; Wang, D.; Jiang, D.; Gao, X. Modification of structure and wear resistance of closed-field unbalanced-magnetron sputtered MoS2 film by vacuum-heat-treatment. Surf. Coat. Technol. 2020, 401, 126215.)

Round 2
Reviewer 2 Report
The manuscript contains full of contradictions between data. The figures have no captions. The definition of dpa is totally wrong. dpa doesn't all contribute to damage(ea. vacancy). Therefore, we can not estimate the dpa directly from vacancies. I cannot find improvements in the manuscript, therefore reject for publication in nanomaterials.
Author Response
Thank you very much for your suggestions and questions.
- The manuscript contains full of contradictions between data.
In this article, we studied the improvement of the irradiation resistance of amorphous MoS2 films by thermal annealing. This study found that the irradiation hardening phenomenon of amorphous MoS2 film is obvious after irradiation, and the hardness after irradiation is more than three times that of the original film, which lead to the embrittlement of the film, which is not conducive to its lubricating properties. This is evident in the results of the friction test.
After thermal annealing, the crystallinity of the film increases. Although this improves the hardness of the film, the amount of change is small. This is far less obvious than the irradiation hardening caused by the point defects introduced when the MoS2 crystal is destroyed by incident ions.
Compared with the original amorphous film, the annealed film shows better resistance to irradiation hardening. In addition, according to the stability of the (002) crystal structure inside the film (providing better lubricity and reflecting the structure's irradiation resistance), it can be found that the film annealed at 300 °C is the best. The friction test results also confirmed this result.
In this study, we can find that the structure-properties of the material are related.
- The figures have no captions.
Thanks for your suggestion, we also found this problem. We have modified some pictures and its description, and hope to meet your requirements.
- The definition of dpa is totally wrong. dpa doesn't all contribute to damage (ea. vacancy). Therefore, we can not estimate the dpa directly from vacancies.
Sorry, there is an error in our statement here. We have made changes to the content of the article as follows:
“Before the irradiation experiment, the irradiation damage of the film was simulated by SRIM, so as to obtain the irradiation fluence corresponding to the irradiation damage level [29]. The films were irradiated with 2 MeV Au2+ ions at ion fluence of 4.34 × 1014 and 1.09 × 1015 ion/cm2 which corresponding to displacement per atom (dpa) of 2 and 5, respectively. The relationship between the fluence of incident Au ion and displacement per atom were correlated by the following formula [30]:
(Please refer to the attached formula) (1)
Ф: Implantation fluence, ion/cm2
Ndisplacement: Number of displacement, 107/(ion·cm)
NW: Tungsten atomic density, atom/cm3 ”
([29] Ziegler J.F.; Ziegler M.D.; Biersack J.P. SRIM-the stopping and range of ions in matter. Nucl. Instrum. Methods Phys. Res. B 2010, 286, 1818-1823.
[30] Ogorodnikova O.V.; Tyburska B.; Alimov V.Kh.; Ertl K. The influence of radiation damage on the plasma-induced deuterium retention in self-implanted tungsten. J. Nucl. Mater. 2011, 415, S661-S666.)

Reviewer 3 Report
The authors answered all the questions posed, the article can be accepted for publication.
Author Response
Thank you very much for approving our research work.